# Prediction of iodine-125 seed implantation efficacy in lung cancer using an enhanced CT-based nomogram model

Deng Guibin[1], Shen Xiaolan[1], Zhang Wei[2], Lan Xiaoli[3], Dehui Liu[1]*

1 The First College of Clinical Medical Science, China Three Gorges University, Yichang Central People's Hospital, Yichang, China, 2 Yichang Hospital of Traditional Chinese Medicine, Yichang, China, 3 Hubei Province Key Laboratory of Molecular Imaging, Wuhan, China

* liudehui8000@163.com

**Data Availability Statement:** In accordance with PLOS's data policy requirements, we have changed the data access contact point to a non-author

## Abstract

### Background

Lung cancer, a leading cause of death, sees variable outcomes with iodine-125 seed implantation. Predictive tools are lacking, complicating clinical decisions. This study integrates radiomics and clinical features to develop a predictive model, advancing personalized treatment.

### Objective

To construct a nomogram model combining enhanced CT image features and general clinical characteristics to evaluate the efficacy of radioactive iodine-125 seed implantation in lung cancer treatment.

### Methods

Patients who underwent lung iodine-125 seed implantation at the Nuclear Medicine Department of Xiling Campus, Yichang Central People's Hospital from January 1, 2018, to January 31, 2024, were randomly divided into a training set (73 cases) and a test set (31 cases). Radiomic features were extracted from the enhanced CT images, and optimal clinical factors were analyzed to construct clinical, radiomics, and combined models. The best model was selected and validated for its role in assessing the efficacy of iodine-125 seed implantation in lung cancer patients.

### Results

Three clinical features and five significant radiomic features were successfully selected, and a combined nomogram model was constructed to evaluate the efficacy of iodine-125 seed implantation in lung cancer patients. The AUC values of the model in the training and test sets were 0.95 (95% CI: 0.91–0.99) and 0.83 (95% CI: 0.69–0.98), respectively. The calibration curve demonstrated good agreement between predicted and observed values, and the

institutional contact. The data can be accessed through the Ethics Committee of our institution, with the following contact details: [Yichang Central People's Hospital Ethics Committee, shenxl2022@163.com]. We have updated the data availability statement in the manuscript and have already sent this information to you via email.

**Funding:** the Open Fund of Hubei Provincial Key Laboratory of Molecular Imaging (No. 2023fzyx024) and the Hubei Provincial Traditional Chinese Medicine Research Project (No. ZY2023M038).

**Competing interests:** The authors have declared that no competing interests exist.

decision curve indicated that the combined model outperformed the clinical or radiomics model across the majority of threshold ranges.

## Conclusion

A combined nomogram model was successfully developed to assess the efficacy of iodine-125 seed implantation in lung cancer patients, demonstrating good clinical predictive performance and high clinical value.

## 1. Introduction

Lung cancer is one of the most common malignant tumors, with primary clinical symptoms including cough, hemoptysis, and chest pain [1–3]. According to the 2020 Global Cancer Observatory (GLOBOCAN) database, there are approximately 2 million new cases and 1.76 million deaths from lung cancer annually. China accounts for 37% of global lung cancer incidence and 39.8% of related mortality. The age of onset is trending younger, and the majority of patients are diagnosed at an advanced stage, significantly affecting treatment outcomes and prognosis [4–6]. Smoking, environmental pollution, and a history of chronic lung diseases are major risk factors for lung cancer, with smoking being the primary factor influencing its incidence [7, 8]. Currently, the clinical treatment of lung cancer depends on the type, stage, and overall health of the patient. The main treatment modalities include surgery, radiotherapy, chemotherapy, targeted therapy, and immunotherapy [9, 10]. However, current treatment strategies, while delaying disease progression, are associated with significant adverse effects, considerable impact on surrounding healthy tissues, and a high recurrence rate, which limit their effectiveness [11].

In recent years, the technology of radioactive iodine-125 seed implantation has developed rapidly. By directly implanting radioactive iodine-125 seeds into the diseased tissue to selectively irradiate tumor cells, a high dose of radiation can be delivered to the affected area. The gamma rays emitted during the decay process can effectively kill tumor cells, making this technique crucial in the local treatment of malignant tumors. Additionally, it causes minimal radiation damage to surrounding healthy tissues, resulting in fewer side effects compared to conventional whole-body radiotherapy [12, 13]. Additionally, iodine-125 seeds release energy gradually within the tumor tissue, allowing for continuous and prolonged treatment of the tumor [14, 15]. Studies have found that iodine-125 seed implantation is effective in the treatment of advanced lung cancer and can counteract the incidence of adverse effects caused by external radiotherapy, chemotherapy, and immunotherapy, thereby improving the patient's quality of life [16, 17]. However, the poor accuracy and uniformity of iodine-125 seed implantation pose challenges in precisely predicting the treatment efficacy and optimizing the risk-benefit ratio.

Radiomics involves extracting information from biomedical images and using diverse statistical analyses and data integration to interpret quantitative features. This approach aids in disease diagnosis, prognosis, personalized treatment, and biomarker discovery, offering advantages such as non-invasiveness, reproducibility, lower cost, and reduced sensitivity to tumor heterogeneity [18–20]. Currently, radiomics plays a crucial role in predicting patient survival, assessing treatment efficacy, identifying tumor molecular subtypes, and discovering tumor biomarkers in oncological diseases. Michele Avanzo and colleagues have found that radiomic models based on computed tomography (CT) and positron emission tomography (PET) have

advanced to the level where they can detect lung nodules, differentiate between malignant and benign lesions, characterize tumors, stage them, and determine genotypes [21]. However, there are currently few reports on using CT radiomics to predict the efficacy of iodine-125 seed implantation for lung cancer. Therefore, the aim of this study is to use radiomics to analyze pre-treatment enhanced CT images of lung cancer patients who will undergo iodine-125 seed implantation. The study involved extracting radiomic features and developed imaging, clinical, and combined models to predict and assess treatment efficacy before implantation, with the goal of providing stronger support for clinical decision-making.

## 2. Materials and methods

### 2.1 Study design

This study was designed as a retrospective analysis aimed at evaluating the efficacy of iodine-125 seed implantation for the treatment of lung cancer. The subjects of the study were lung cancer patients treated at the Department of Nuclear Medicine, Xiling District, Yichang Central People's Hospital, between January 1, 2018, and January 31, 2024.

 The study was conducted at the Department of Nuclear Medicine, Xiling District, Yichang Central People's Hospital, a tertiary care hospital providing comprehensive treatment for lung cancer patients in the region.

### 2.2 Study subjects

The inclusion criteria were as follows: Patients who (i)had pathological staging of III or higher. (ii)were clinically diagnosed lung cancer undergoing preoperative enhanced chest CT within one week. (iii) had not received external radiation therapy or chemotherapy concurrently. (iv) were receiving treatment for the first time at our department. (v) were unable or unwilling to undergo conventional surgical treatment. (vi)had a tumor diameter less than 7 cm.

 The exclusion criteria were as follows: Patients who (i)had bleeding disorders. (ii)had uncontrolled inflammation around the lesion. (iii)had liver, kidney, or heart dysfunction.

 This study was a retrospective analysis conducted in May 2024, adhering to the principles of the Declaration of Helsinki. The study received written approval from the Ethics Committee of Yichang Central People's Hospital (Approval No: 2023-248-01). Given the retrospective nature of this study, the Ethics Committee waived the requirement for obtaining informed consent from patients, as all data were fully anonymized before access. A total of 104 patients were included, Patients were randomly divided into a training set (n = 73) and a testing set (n = 31) in a 7:3 ratio [22, 23].

### 2.3 Efficacy assessment

Efficacy was evaluated based on the RECIST guidelines (version 1.1) for solid tumors. Three months after lung particle implantation, efficacy was assessed according to follow-up CT results. According to the guidelines, results were classified into complete response (CR), partial response (PR), stable disease (SD), and progressive disease (PD). Patients with CR and PR were classified as positive, while those with SD and PD were classified as negative [24].

### 2.4 CT image acquisition and preprocessing

 **2.4.1 CT image acquisition.**  All patients underwent enhanced chest CT scans within one week before treatment. Scanning was performed using a SOMATOM Definition Flash scanner (Siemens). A high-pressure injector was used to administer iodixanol intravenously via the antecubital vein, with a concentration of 350 mg/mL, a dose of 1.5–2 mL/kg, and an injection

rate of 3 mL/s. The arterial phase scan was initiated after a 25-second delay. Scanning parameters included a tube voltage of 120 kV, a tube current of 110 mAs, a pitch of 0.2, a conventional slice thickness of 5.0 mm, and standard algorithm reconstruction with a reconstructed slice thickness of 0.75 mm. The scan results were exported in DICOM format. Image preprocessing involved normalization to reduce differences between image acquisition protocols. First, images were resampled to a size of 1×1×1 mm, and then a bin width of 25 was set for gray-level discretization.

**2.4.2 Regions of interest delineation and feature extraction.** The enhanced CT lung window images of 104 patients were sequentially imported into ITK-SNAP (V4.0.1) software. All lung cancer regions of interest (ROI) were independently delineated by a radiologist with over five years of experience. Necrotic areas, intraluminal air, trachea, and blood vessels surrounding the tumor were excluded from the ROI during delineation. Another radiologist with over ten years of experience randomly selected 15 cases from the 104 patients for independent ROI delineation. Radiomic features were batch-extracted from the delineated ROIs using the Pyradiomics (V3.0.1) package in Python. Features with an interclass correlation coefficient (ICC) greater than 0.8 between the two radiologists were included in the final feature selection. All features were standardized using the Z-score to minimize the influence of extreme values. The radiomic features were categorized into eight sets: Original, Exponential, Gradient, LBP2D, Logarithm, Square, SquareRoot, and Wavelet. Each category included first-order features, gray-level co-occurrence matrix (GLCM) features, gray-level size zone matrix (GLSZM) features, gray-level run length matrix (GLRLM) features, gray-level dependence matrix (GLDM) features, and neighboring gray tone difference matrix (NGTDM) features.

## 2.5 Feature selection and model construction

The least absolute shrinkage and selection operator (LASSO), support vector machine (SVM), and random forest (RF) algorithms were used to select an equal number of the most representative non-zero coefficient features from all radiomic features in the training set. A feature selection model was established using binary logistic regression, and the accuracy, sensitivity, area under the curve (AUC) and other performance evaluation indexes of the training and test sets were calculated to evaluate the model's performance. The feature selection method with the best performance was chosen to build the radiomics model based on the selected features. The Radscore was calculated based on the weight coefficients of the optimal features: Radscore $= \beta_1 X_1 + \beta_2 X_2 + \beta_3 X_3 + \ldots + \beta_n X_n$, where $X_n$ represents the expression level of the radiomic feature and $\beta_n$ is the feature weight. The Radscore's statistical significance in predicting treatment efficacy was also evaluated. Univariate logistic regression was used to select clinical features with a P-value $< 0.1$, which were then included in a multivariate logistic regression analysis to identify the best clinical features and construct a clinical model. A combined model was established by integrating the Radscore and clinical features. The workflow is illustrated in Fig 1.

## 2.6 Statistical methods

Statistical analysis was performed using R Studio (2023.06.1+524) and SPSS (19.0) software. Continuous variables were expressed as mean ± standard deviation. An independent sample T-test was used for comparisons if the data followed a normal distribution, while the Mann-Whitney U test was used for non-normally distributed data. Categorical variables were expressed as frequency (percentage). Receiver operating characteristic (ROC) curves, calibration curves, and decision curves were plotted to assess the discriminatory power of the models. The best-performing model was visualized using a nomogram. The Delong test was used to

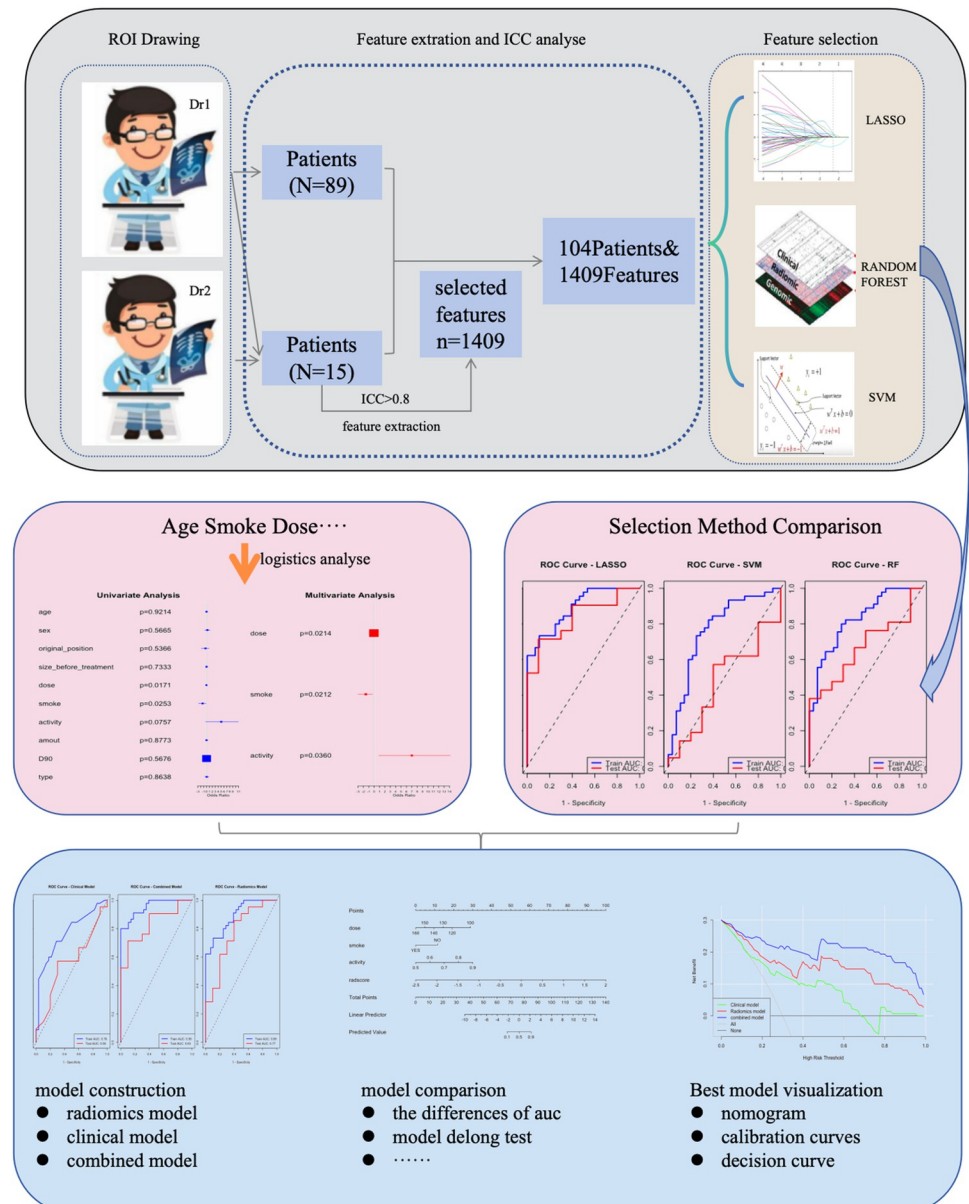

**Fig 1. Flow chart.**

compare the AUCs of different models, with a P-value < 0.05 indicating statistical significance.

## 3. Results

### 3.1 General data analysis

The baseline characteristics of 104 patients who underwent Iodine-125 seed implantation for lung cancer are summarized in Table 1. In the training set, 45 patients were positive and 28 were negative. In the test set, 21 patients were positive and 10 were negative. Among the baseline characteristics in the training set, Dose and Smoke variables showed statistically significant differences, with *P*-values of 0.01 and 0.02, respectively.

**Table 1. Clinical data of patients in training set and test set ($\bar{X} \pm s$ & [n(%)]).**

| Variable | Non-responder (n = 28) | Responder (n = 45) | P | Non-responder (n = 10) | Responder (n = 21) | P |
|---|---|---|---|---|---|---|
| Year | 63.75 ± 10.50 | 64.00 ± 10.78 | 0.923 | 61.40 ± 6.40 | 65.57 ± 11.49 | 0.295 |
| Tumor Size(mm) | 62.72 ± 42.42 | 66.25 ± 44.25 | 0.737 | 53.20 ± 33.05 | 53.92 ± 23.84 | 0.945 |
| Dose(Gy) | 138.21 ± 11.24 | 130.00 ± 14.30 | 0.012 | 142.00 ± 10.33 | 133.81 ± 11.17 | 0.061 |
| Number of particles | 109.46 ± 46.13 | 111.40 ± 56.48 | 0.879 | 83.70 ± 63.17 | 96.24 ± 41.17 | 0.511 |
| Postoperative D90(Gy) | 169.30 ± 56.90 | 234.81 ± 44.66 | 0.444 | 189.5 1± 64.02 | 188.56 ± 82.64 | 0.974 |
| Gender | | | 0.566 | | | 1.000 |
| women | 15 (53.57) | 21 (46.67) | | 2 (20.00) | 6 (28.57) | |
| men | 13 (46.43) | 24 (53.33) | | 8 (80.00) | 15 (71.43) | |
| Primary location | | | 0.760 | | | 0.087 |
| Other | 4 (14.29) | 9 (20.00) | | 3 (30.00) | 1 (4.76) | |
| Lung-derived | 24 (85.71) | 36 (80.00) | | 7 (70.00) | 20 (95.24) | |
| Smoke | | | 0.020 | | | 0.447 |
| No | 4 (14.29) | 18 (40.00) | | 5 (50.00) | 7 (33.33) | |
| Yes | 24 (85.71) | 27 (60.00) | | 5 (50.00) | 14 (66.67) | |
| Particle Activity(mci) | | | 0.240 | | | 0.686 |
| 0.5 | 5 (17.86) | 4 (8.89) | | 2 (20.00) | 2 (9.52) | |
| 0.6 | 13 (46.43) | 14 (31.11) | | 5 (50.00) | 10 (47.62) | |
| 0.7 | 8 (28.57) | 20 (44.44) | | 1 (10.00) | 6 (28.57) | |
| 0.8 | 1 (3.57) | 6 (13.33) | | 2 (20.00) | 2 (9.52) | |
| 0.9 | 1 (3.57) | 1 (2.22) | | 0 (0.00) | 1 (4.76) | |
| Tumor Type | | | 0.786 | | | 0.440 |
| Squamous Cell Carcinoma | 10 (35.71) | 17 (37.78) | | 5 (50.00) | 9 (42.86) | |
| Adenocarcinoma | 15 (53.57) | 21 (46.67) | | 5 (50.00) | 8 (38.10) | |
| Metastatic Cancer | 3 (10.71) | 7 (15.56) | | 0 (0.00) | 4 (19.05) | |

## 3.2 Feature selection and radscore construction

A total of 1,569 radiomic features were extracted from each ROI. Among the 15 patients whose ROIs were delineated by both physicians, 1,409 features had an ICC > 0.8, which were then used as the final features for all patients. The LASSO, SVM, and RF methods were applied to select features from the 1,409 features in the training set, with each method identifying 5 non-zero coefficient features most closely related to the efficacy of seed implantation. The results indicated that the features selected by LASSO had significantly higher accuracy, sensitivity, and AUC values, etc. in both the training and test sets compared to the other two methods. The AUC values for LASSO were 0.89 (95% CI: 0.82–0.96) in the training set and 0.83 (95% CI: 0.67–0.98) in the test set (Table 2 and Fig 2A). The Radscore was calculated based on

**Table 2. Performance indicators of different feature screening methods.**

| Method | DataSet | Accuracy | Sensitivity | Specificity | Precision | F1 | AUC(95%CI) |
|---|---|---|---|---|---|---|---|
| LASSO | Train | 0.75 | 0.61 | 0.84 | 0.71 | 0.65 | 0.89(0.82–0.96) |
| | Test | 0.84 | 0.67 | 0.91 | 0.75 | 0.71 | 0.83(0.67–0.98) |
| SVM | Train | 0.74 | 0.57 | 0.84 | 0.70 | 0.63 | 0.76(0.65–0.88) |
| | Test | 0.52 | 0.40 | 0.57 | 0.31 | 0.35 | 0.47(0.25–0.69) |
| RF | Train | 0.74 | 0.54 | 0.87 | 0.71 | 0.61 | 0.82(0.73–0.92) |
| | Test | 0.61 | 0.30 | 0.76 | 0.38 | 0.33 | 0.66(0.47–0.86) |

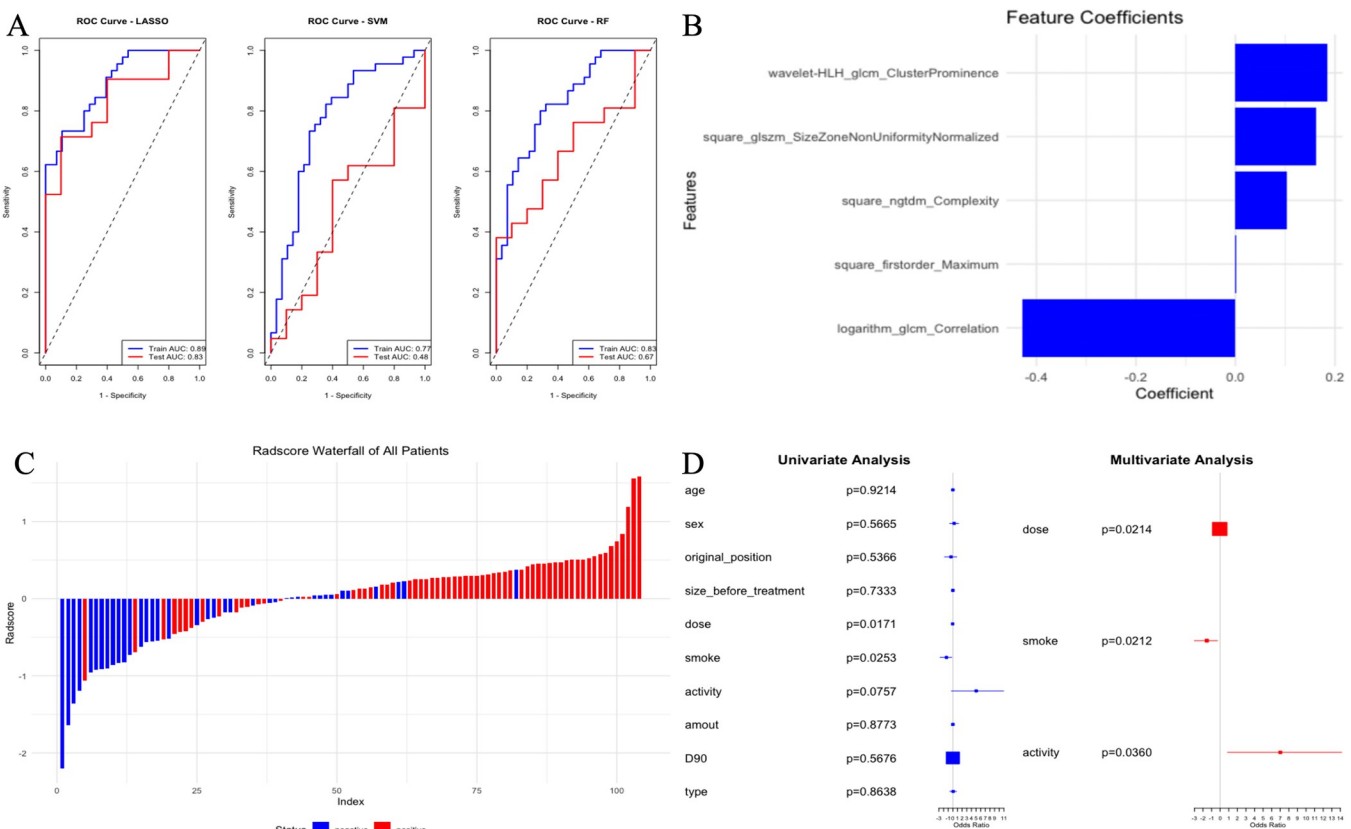

**Fig 2. Screening of imaging and clinical features.** A: ROC curves of LASSO, SVM and RF screening methods in training set and test set; B: LASSO filters 5 feature names and feature weights; C: Radscore values of all patients; D: Clinical features first single and then multi-factor logistic regression analyse.

the expression levels and weights of the 5 features selected by LASSO (Fig 2B). The Radscores of all patients are shown in Fig 2C, with positive having significantly higher Radscores than negative in both sets, indicating a statistically significant difference in clinical efficacy ($P<0.01$; $P = 0.01$).

### 3.3 Model construction and validation

Univariate and subsequent multivariate logistic regression analyses were performed on the clinical features of the training set, with a threshold of $P<0.1$ for univariate analysis and $P<0.05$ for multivariate analysis. The results identified seed dose, seed activity, and smoking status as independent predictors of seed implantation efficacy (Fig 2D). The clinical model constructed based on these predictors had AUC values of 0.76 (95% CI: 0.65–0.87) in the training set and 0.56 (95% CI: 0.35–0.78) in the test set. The radiomics model, constructed using the 5 features selected by LASSO, had AUC values of 0.89 (95% CI: 0.82–0.96) in the training set and 0.77 (95% CI: 0.58–0.95) in the test set. The combined model, which integrated seed dose, seed activity, smoking status, and Radscore, achieved AUC values of 0.95 (95% CI: 0.91–0.99) in the training set and 0.83 (95% CI: 0.69–0.98) in the test set.

Comparing the accuracy, sensitivity, and AUC values, etc. of the three models showed that the combined model outperformed the other two models in all metrics (Table 3 and Fig 3A). Delong's test demonstrated that the difference in AUC between the combined model and the clinical model was statistically significant in both the training and test sets ($Z = -3.56$, $P<0.01$;

**Table 3. Different model performance indicators in training set and test set.**

| Model | Dataset | Accuracy | Sensitivity | Specificity | Precision | F1 | AUC(95%CI) |
|---|---|---|---|---|---|---|---|
| Clinical | Train | 0.71 | 0.50 | 0.84 | 0.67 | 0.57 | 0.76(0.65–0.87) |
| | Test | 0.58 | 0.40 | 0.67 | 0.36 | 0.38 | 0.56(0.35–0.78) |
| Combined | Train | 0.86 | 0.82 | 0.89 | 0.82 | 0.82 | 0.95(0.91–0.99) |
| | Test | 0.81 | 0.60 | 0.90 | 0.75 | 0.67 | 0.83(0.69–0.98) |
| Radiomics | Train | 0.75 | 0.61 | 0.84 | 0.71 | 0.65 | 0.89(0.82–0.96) |
| | Test | 0.71 | 0.60 | 0.76 | 0.55 | 0.57 | 0.77(0.58–0.95) |

$Z$ = -3.09, $P<0.01$). The difference between the combined model and the radiomics model was statistically significant in the training set ($Z$ = -2.17, $P$ = 0.03) but not in the test set ($Z$ = 1.21, $P$ = 0.23). The difference between the radiomics model and the clinical model was not statistically significant ($Z$ = -1.11, $P>0.05$). The combined model was visualized as a nomogram (Fig 3B). Calibration curves (Fig 3C and 3D) indicated good calibration in both the training and test sets ($\chi^2$ = 7.95, $P$ = 0.44; $\chi^2$ = 3.58, $P$ = 0.89). Decision curve analysis showed that the combined model had the highest net benefit across most high-risk thresholds, outperforming both the clinical and radiomics models alone (Fig 3E).

## 4. Discussion

### 4.1 Iodine-125 seed implantation for lung cancer

Lung cancer is a leading cause of cancer-related deaths worldwide [25], and its treatment at advanced stages relies heavily on chemoradiotherapy and immunotherapy [26, 27]. However,

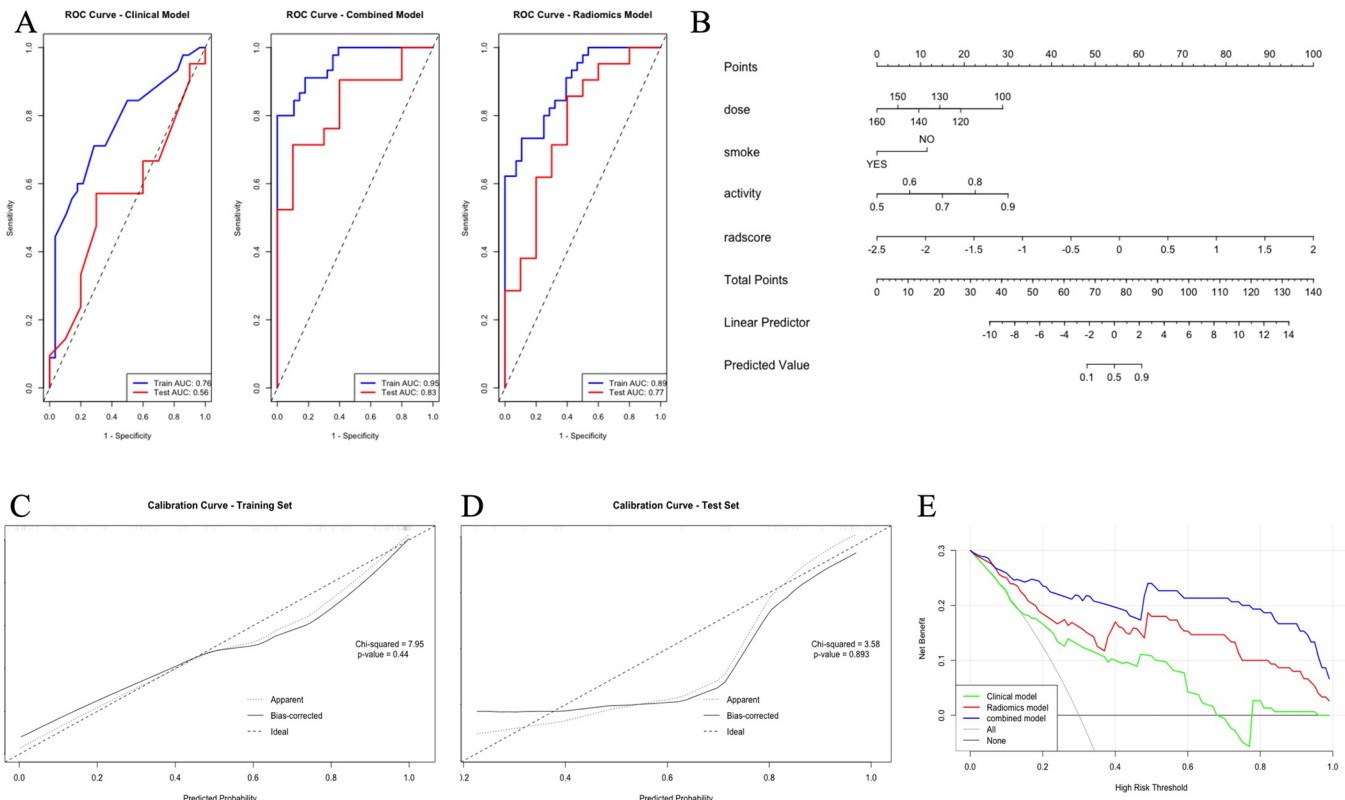

**Fig 3. Model construction and validation.** A: ROC curves of clinical, imaging and joint models in training and test sets; B: Joint model nomogram model; C: Training set calibration curve; D: test set calibration curve E: decision curve.

traditional treatments pose challenges such as damage to healthy tissues and high recurrence rates [11]. Iodine-125 seed implantation, as an internal radiation therapy, provides prolonged targeted irradiation and has become increasingly used in clinical practice. While this method has shown promise in slowing tumor growth, research on predicting the efficacy of iodine-125 seed implantation based on preoperative CT remains limited [28, 29]. Therefore, this study aims to explore the predictive value of combining radiomic features with clinical characteristics in assessing the efficacy of iodine-125 seed implantation for lung cancer.

## 4.2 Advantages of radiomics in iodine-125 treatment for lung cancer

In recent years, radiomics has made significant advancements in oncology and has been widely applied in tumor diagnosis, staging, prognostic assessment, and treatment response prediction. Additionally, research indicates that iodine-125 seed implantation markedly improves both short-term and long-term outcomes for cancer patients [30, 31]. In this study, five features closely associated with the efficacy of iodine-125 seed implantation were selected from a total of 1569 imaging features. Notably, the feature "glcm_ClusterProminence" was identified as a common feature by both LASSO and SVM methods. A high value of this feature may indicate tumor tissue heterogeneity, suggesting a complex cellular structure within the region. This high expression is closely related to poor prognosis, such as suboptimal treatment response, indicating that this feature has significant diagnostic and prognostic value [32]. Additionally, the analysis of clinical features revealed that particle dose, smoking status, and particle activity are independent predictors of treatment efficacy, which is consistent with previous studies. Iodine-125 seeds primarily release low-energy γ-rays, causing DNA strand breaks in tumor cells [33], a higher seeds activity can deliver a sufficient radiation dose in a shorter period, and the greater the radiation dose received by tumor cells, the more severe the DNA damage. This leads to a decrease in cellular proliferation and may even induce apoptosis. Therefore, the combined effect of seeds dose and activity directly influences the postoperative efficacy in patients [34, 35]. Smoking impacts the efficacy of iodine-125 seed therapy through various mechanisms, including damaging lung function, reducing blood oxygen levels, and suppressing the immune system. These effects contribute to decreased treatment effectiveness [36, 37]. The Radscore was calculated using the expression values and weights of the five imaging features and was then combined with three clinical factors to construct a joint model. However, the clinical model, based on clinical features alone, and the radiomics model, based on imaging features alone, both demonstrated significantly lower accuracy, sensitivity, specificity, and AUC values in both the training and testing sets compared to the joint model. Delong's test confirmed that the AUC differences between the combined model and two other models were statistically significant ($P<0.01$). This indicates that the joint model provides a more comprehensive prediction of treatment efficacy for lung cancer patients undergoing iodine-125 seed implantation. While clinical features offer individualized information about the patient's overall health, treatment history, and lifestyle, radiomics precisely reflects the tumor's physical morphology and texture. By integrating both, the model not only captures internal tumor changes but also accounts for individual patient differences, enhancing the model's stability and generalizability. This explains why the joint model outperforms the other two models.

## 4.3 Innovations and limitations

This study represents a significant advancement by integrating radiomic features and clinical characteristics to assess the efficacy of iodine-125 seed implantation in lung cancer. Previous research has often focused on single types of features; however, this study demonstrates the

enhanced predictive capability achieved through combining multidimensional features via feature selection and model construction. This approach not only provides new insights for the clinical application of iodine-125 seed implantation in lung cancer but also offers practical references for future research integrating radiomics with clinical features.

Despite the meaningful results achieved in predicting lung cancer treatment efficacy using a combination of radiomic and clinical features, several limitations persist. First, the study is a single-center, retrospective analysis with a relatively small sample size, which may lead to selection bias. Larger, multicenter studies are needed for further validation. Second, future research could incorporate MRI or PET to provide more comprehensive tumor detail. Lastly, extending the follow-up period would allow for a more thorough evaluation of the long-term effects of iodine-125 seed implantation and facilitate comparisons with other treatment methods.

## 5 Conclusion

Theoretically, this study supports the effectiveness of radiomics as a tool for predicting tumor treatment efficacy while highlighting the importance of clinical characteristics in personalized medicine. Future research should further explore the integration of radiomics with other types of data, such as genomics and proteomics, to develop more comprehensive and precise predictive models.

In summary, this research provides new perspectives on predicting the efficacy of iodine-125 seed implantation for lung cancer treatment and lays the groundwork for future applications of radiomics in oncology. With advancements in technology and further research, the use of radiomics in personalized medicine holds promise for improving patient survival rates and quality of life.

## Acknowledgments

Thanks for the cooperation of all authors and the support of various units and departments.

## Author Contributions

**Data curation:** Deng Guibin, Shen Xiaolan.

**Formal analysis:** Zhang Wei.

**Funding acquisition:** Zhang Wei, Lan Xiaoli, Dehui Liu.

**Methodology:** Lan Xiaoli, Dehui Liu.

**Software:** Deng Guibin, Lan Xiaoli.

**Supervision:** Zhang Wei, Dehui Liu.

**Validation:** Shen Xiaolan, Lan Xiaoli.

**Writing – original draft:** Deng Guibin, Shen Xiaolan.

**Writing – review & editing:** Zhang Wei, Lan Xiaoli, Dehui Liu.

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
