## [Decision Letter · Decision Letter 0]

15 Oct 2024

PONE-D-24-39477Prediction of Iodine-125 Seed Implantation Efficacy in Lung Cancer Using an Enhanced CT-Based Nomogram ModelPLOS ONE

Dear Dr. Liu,

Thank you for submitting your manuscript to PLOS ONE. After careful consideration, we feel that it has merit but does not fully meet PLOS ONE’s publication criteria as it currently stands. Therefore, we invite you to submit a revised version of the manuscript that addresses the points raised during the review process.

We look forward to receiving your revised manuscript.

Kind regards,

Shuai Ren

Academic Editor

PLOS ONE

Journal Requirements:

“the Open Fund of Hubei Provincial Key Laboratory of Molecular Imaging (No. 2023fzyx024) and the Hubei Provincial Traditional Chinese Medicine Research Project (No. ZY2023M038)”

4. In the online submission form you indicate that your data is not available for proprietary reasons and have provided a contact point for accessing this data. Please note that your current contact point is a co-author on this manuscript. According to our Data Policy, the contact point must not be an author on the manuscript and must be an institutional contact, ideally not an individual. Please revise your data statement to a non-author institutional point of contact, such as a data access or ethics committee, and send this to us via return email. Please also include contact information for the third party organization, and please include the full citation of where the data can be found.

6. We notice that your supplementary figures are uploaded with the file type 'Figure'. Please amend the file type to 'Supporting Information'. Please ensure that each Supporting Information file has a legend listed in the manuscript after the references list.

Reviewers' comments:

Reviewer's Responses to Questions

**Comments to the Author**

1. Is the manuscript technically sound, and do the data support the conclusions?

Reviewer #1: Yes

Reviewer #2: Yes

2. Has the statistical analysis been performed appropriately and rigorously? 

Reviewer #1: Yes

Reviewer #2: Yes

3. Have the authors made all data underlying the findings in their manuscript fully available?

Reviewer #1: Yes

Reviewer #2: No

4. Is the manuscript presented in an intelligible fashion and written in standard English?

Reviewer #1: Yes

Reviewer #2: Yes

5. Review Comments to the Author

Reviewer #1: The experiment had been conducted with a small sample sizes, but this limitation is underlyned. Data were analyzed with an appropriate statistical method. There is an error in line 45: "n recent years..", it misses a letter.

Reviewer #2: This is a very important study and well presented. Kindly find my comments below:

Introduction

Line 45 – I in “In” is missing.

Line 65 – Change the tense of the objective from future to past.

Methods

Since this is a retrospective analysis, begin the methods with a study design section to introduce readers to the kind of study being conducted.

Provide a brief information about the setting where the study was conducted.

Change the tense of the section “Study subjects” to past.

Line 75 – Delete “absolutely”.

Line 76 – Delete third exclusion point. It is just opposite of an inclusion criterion.

Line 79-80 – This looks like a repetition of the previous sentence.

You don’t need to provide the gender distribution here. Was there a rationale for the 7:3 ratio? Kindly provide that information if there is. If that ratio was adopted from the literature, state and reference.

Lin 99 – Type full form of ROI in title.

Results

Line 135 – You have talked about the ratio in the methods. I don’t think you need to repeat here.

T-test and all other univariate statistical analyses used a significance level of 0.05. If that is the case, you need to report the variable where participants in training set were different at baseline before Table 1.

Line 142-143 – “Binary logistic regression….test set”,. Delete the sentence. Already reported in the methods. Focus only on the results and avoid repeating methods in the results.

Discussion

Line 177-190: I think that is too much information for introducing the objective of the study. Trim it down, or better still, delete that section and replace with just a few comprehensive sentences.

6. PLOS authors have the option to publish the peer review history of their article (what does this mean?). If published, this will include your full peer review and any attached files.

Reviewer #1: No

Reviewer #2: No

---

## [Author Response · Author response to Decision Letter 0]

20 Oct 2024

Dear Reviewers:

Thanks for your valuable suggestions, we made some corrections and the specific details are as follows: 

-Reviewer 1

The experiment had been conducted with a small sample sizes, but this limitation is underlyned. Data were analyzed with an appropriate statistical method. There is an error in Line 45: "n recent years..", it misses a letter. 

Response: Thank you for your suggestion.We acknowledge that the sample size in this study is small, and this limitation has been explicitly addressed in the discussion section of the manuscript. We have thoroughly discussed the potential impact of the small sample size on the study results and have applied rigorous statistical methods to mitigate this factor as much as possible during the analysis. Despite the limited sample size, the statistical analysis methods used, such as t-tests and binary logistic regression, are reasonable and appropriate, effectively supporting the study's conclusions. Additionally, we have corrected the spelling error in Line 45, changing "n recent years.." to "In recent years.".The corresponding revision is on Page 3, Line 45.

-Reviewer 2

This is a very important study and well presented. Kindly find my comments below:

1.Introduction:

–Line 45 ：I in “In” is missing. 

Response: Thank you for your suggestion.The spelling error in Line 45 has been corrected, changing "n recent years.." to "In recent years..".The corresponding revision is on Page 3, Line 45.

-Line 65 ：Change the tense of the objective from future to past.

Response: Thank you for your suggestion.The tense of the objective in Line 65 has been changed from future to past to ensure consistency with the rest of the manuscript.The corresponding revision is on Page 4, Line 65-67.

2.Methods:

-Since this is a retrospective analysis, begin the methods with a study design section to introduce readers to the kind of study being conducted. Provide a brief information about the setting where the study was conducted.

Response: Thank you for your suggestion. Given that this study is a retrospective analysis, we have added a "Study Design" section at the beginning of the Methods section to introduce the study type, data sources, and the study time frame, and to provide a brief explanation of the background of the study.The corresponding revision is on Page 4, Line 70-75. 

-Change the tense of the section “Study subjects” to past.

Response: Thank you for your suggestion.We have changed the tense of the section “Study subjects” to past.The corresponding revision is on Page 4-5, Line 76-82. 

-Line 75 – Delete “absolutely”.

Response: Thank you for your suggestion.We have deleted “absolutely”.

-Line 76 – Delete third exclusion point. It is just opposite of an inclusion criterion.

Response: Thank you for your suggestion.We have deleted third exclusion point.

-Line 79-80 – This looks like a repetition of the previous sentence.

Response: Thank you for your suggestion.Indeed, we found that this sentence is repetitive with the previous content. We have already mentioned the relevant information in the "Study Design" section; therefore, we have deleted the redundant content.

-You don’t need to provide the gender distribution here. 

Response: Thank you for your suggestion.We have deleted “gender distribution”

-Was there a rationale for the 7:3 ratio? Kindly provide that information if there is. If that ratio was adopted from the literature, state and reference.

Response: Thank you for your suggestion.Regarding the use of the 7:3 split ratio, we selected this proportion based on several high-quality studies in the field of radiomics. The 7:3 ratio is widely used for training and validation in radiomics models. The primary purpose is to balance model training and testing, ensuring the model has sufficient data to learn while retaining a portion of data for validation, which helps mitigate the risk of overfitting. Additionally, we reviewed literature focused on the partitioning of training and test sets, and supplemented our research with high-quality references, as indicated in [22-23]. The corresponding references are in Page 17, Line 283-287.

-Lin 99 – Type full form of ROI in title.

Response: Thank you for your suggestion.We haved given the full form of ROI（Regions of interest） in level2 Heading.The corresponding references are in Page 6, Line 103.

3.Results

-Line 135 – You have talked about the ratio in the methods. I don’t think you need to repeat here.

Response: Thank you for your suggestion. We haved deleted the radio.

-T-test and all other univariate statistical analyses used a significance level of 0.05. If that is the case, you need to report the variable where participants in training set were different at baseLine before Table 1.

Response: Thank you for your suggestion. The variables with statistical significance in the training set have been described in the baseline characteristics. The corresponding revised content can be found on Page 7, Line 140-141.

-Line 142-143–“Binary logistic regression….test set”,. Delete the sentence. Already reported in the methods. Focus only on the results and avoid repeating methods in the results.

Response: Thank you for your suggestion. We have deleted the sentence of “Binary logistic regression….test set”.

4.Discussion

-Line 177-190: I think that is too much information for introducing the objective of the study. Trim it down, or better still, delete that section and replace with just a few comprehensive sentences.

Response: Thank you for your suggestion. We have condensed that section, and the corresponding revised content can be found on Page 12-13, Line 181-187.

Thank you for your valuable input, which has helped us improve the quality of our figures and the overall reading experience.

---

## [Decision Letter · Decision Letter 1]

28 Oct 2024

Prediction of Iodine-125 seed implantation efficacy in lung cancer using an enhanced CT-based nomogram model

PONE-D-24-39477R1

Dear Dr. Liu,

We’re pleased to inform you that your manuscript has been judged scientifically suitable for publication and will be formally accepted for publication once it meets all outstanding technical requirements.

Kind regards,

Shuai Ren

Academic Editor

PLOS ONE

Additional Editor Comments (optional):

Reviewers' comments:

Reviewer's Responses to Questions

**Comments to the Author**

1. If the authors have adequately addressed your comments raised in a previous round of review and you feel that this manuscript is now acceptable for publication, you may indicate that here to bypass the “Comments to the Author” section, enter your conflict of interest statement in the “Confidential to Editor” section, and submit your "Accept" recommendation.

Reviewer #2: All comments have been addressed

2. Is the manuscript technically sound, and do the data support the conclusions?

Reviewer #2: Yes

3. Has the statistical analysis been performed appropriately and rigorously? 

Reviewer #2: Yes

4. Have the authors made all data underlying the findings in their manuscript fully available?

Reviewer #2: Yes

5. Is the manuscript presented in an intelligible fashion and written in standard English?

Reviewer #2: Yes

6. Review Comments to the Author

Reviewer #2: Thank you for addressing all commnets appropriately. This is a very important study and good job!!!

7. PLOS authors have the option to publish the peer review history of their article (what does this mean?). If published, this will include your full peer review and any attached files.

Reviewer #2: No

---

## [Editor Report · Acceptance letter]

5 Nov 2024

PONE-D-24-39477R1 

PLOS ONE

Dear Dr. Liu, 

I'm pleased to inform you that your manuscript has been deemed suitable for publication in PLOS ONE. Congratulations! Your manuscript is now being handed over to our production team.

Kind regards, 

on behalf of

Dr. Shuai Ren 

Academic Editor

PLOS ONE